# Effect of Different Post-Sintering Temperatures on the Microstructures and Mechanical Properties of a Pre-Sintered Co–Cr Alloy

**Seong-Ho Jang [1], Bong Ki Min [2], Min-Ho Hong [3],\***  **and Tae-Yub Kwon [3,4],\***

[1] Department of Dental Science, Graduate School, Kyungpook National University, Daegu 41940, Korea; moodttl@knu.ac.kr
[2] Center for Research Facilities, Yeungnam University, Gyeongsan 38541, Korea; bkmin@ynu.ac.kr
[3] Institute for Biomaterials Research & Development, Kyungpook National University, Daegu 41940, Korea
[4] Department of Dental Biomaterials, School of Dentistry, Kyungpook National University, Daegu 41940, Korea
\* Correspondence: mhhong@knu.ac.kr (M.-H.H.); tykwon@knu.ac.kr (T.-Y.K.);
Tel.: +82-53-660-6882 (M.-H.H.); +82-53-660-6891 (T.-Y.K.)

**Abstract:** Although a cobalt–chromium (Co–Cr) blank in a pre-sintered state has been developed, there are few data on the optimal temperature for the alloy in terms of the desired mechanical properties. A metal block (Soft Metal, LHK, Chilgok, Korea) was milled to produce either disc-shaped or dumbbell-shaped specimens. All the milled specimens were post-sintered in a furnace at 1250, 1350 or 1450 °C. The microstructures, shrinkage and density of the three different alloys were investigated using the disc-shaped specimens. The mechanical properties were investigated with a tensile test according to ISO 22674 ($n = 6$). The number and size of the pores in the alloys decreased with increased temperature. The shrinkage and density of the alloys increased with temperature. In the 1250 °C alloy, the formation of the $\varepsilon$ (hexagonal close-packed) phase was more predominant than that of the $\gamma$ (face-centered cubic) phase. The 1350 °C and 1450 °C alloys showed $\gamma$ phase formation more predominantly. Carbide formation was increased along with temperature. The 1450 °C group showed the largest grain size among the three groups. In general, the 1350 °C group exhibited mechanical properties superior to the 1250 °C and 1450 °C groups. These findings suggest that 1350 °C was the most optimal post-sintering temperature for the pre-sintered blank.

**Keywords:** cobalt–chromium alloy; powder metallurgy; sintering; microstructure; mechanical properties

## 1. Introduction

Cobalt–chromium (Co–Cr) dental alloys have been in widespread use for the fabrication of fixed and removable partial dentures, mainly due to their excellent mechanical properties [1,2]. The casting method has traditionally been used to produce Co–Cr-based dental metallic restorations [2,3]. However, making a metallic framework by casting is a prolonged and complex process, leading to multiple complications affecting the final quality [4,5]. In particular, the fabrication of Co–Cr dental prostheses using casting is often difficult, mainly due to the high melting point, high hardness, and low ductility of the alloys [2].

Recent advances in the industry have allowed the fabrication of dental restorations by the use of computer-aided design/computer-aided manufacturing (CAD/CAM) technologies [6]. These digital-based techniques can greatly increase the efficiency of the work time and help prevent operator errors. In these CAD/CAM techniques, Co–Cr-based dental prostheses are produced using either

of the two main techniques: subtractive manufacturing (either hard or soft machining) and additive manufacturing (e.g., selective laser melting) [3,5,7,8]. Although additive manufacturing is rapidly expanding in a number of industrial and dental applications, it is still handicapped by the low productivity of the process, poor quality and uncertain mechanical properties of the final structures [9]. The subtractive process of the hard milling of Co–Cr blanks under standardized industrial conditions minimizes the formation of the flaws and porosities caused by casting [8,10]. However, the high rigidity of the solid blank can cause increased tool and machine wear, ultimately increasing acquisition time and maintenance costs [8,11].

In Co–Cr soft machining, another subtractive manufacturing technique, a green state (un-sintered) blank is post-sintered in a furnace to full density after finishing the milling procedure [2,8], thus overcoming the main disadvantage of hard machining. Moreover, the metallic structures can be relatively easily produced using available CAD/CAM equipment in ordinary dental laboratories [12]. For metallic restorations prepared by soft machining, the alloy shrinkage after post-sintering is approximately 10% [13]. In the soft milling processes, the post-sintering temperature is a key parameter that determines the microstructure and, as a result, the mechanical properties of the final products [14–16].

Commercially available green state Co–Cr blanks include certain organic binders to create the overall cohesion of the metal compacts. In a soft Co–Cr blank, therefore, the powder is homogeneously distributed in a binder, which is capable of burn-out [8]. The binders are removed through thermal debinding during the post-sintering procedure [16]. However, some binder may remain even after most of it is burned out during post-sintering [8]. In addition, the post-sintering step to produce a final metallic structure often generates a lot of smoke. Recently, a Co–Cr blank in pre-sintered state was developed to overcome these disadvantages of green state metallic blanks.

Although the post-sintering temperature may strongly affect the quality of sintered metallic products [8,15], there are few data on the optimal temperature for pre-sintered Co–Cr blanks in terms of the desired mechanical properties. The purpose of this study was to evaluate the effect of three selected post-sintering temperatures on the microstructures and mechanical properties of a pre-sintered Co–Cr alloy. The null hypothesis was that there would be no significant differences in the microstructures and, therefore, in the mechanical properties among the alloys prepared with three different post-sintering temperatures.

## 2. Materials and Methods

### 2.1. Specimen Preparation

A commercial pre-sintered Co–Cr blank (Soft Metal, LHK, Chilgok, Korea), with a composition of Co 63 wt%, Cr 29 wt%, molybdenum 5.8 wt% and trace amounts of silicon, was used. To prepare the blank, Co–Cr powders were mixed with a 10 wt% polymer binder that consisted of polypropylene and stearic acid. The mixture was pressed using a uniaxial pressure (50–65 MPa) to form a disc shape, and the object was then compacted at 300 MPa using cold isostatic pressure. For debinding, the block was heated up to 400 °C with a rate of 5 °C/h. The pre-sintering of the block was performed by heating up to 750 °C with a rate of 5 °C/h, followed by cooling to room temperature.

The pre-sintered metal block was dry-milled using a 5-axis milling machine (T1, Wieland Dental + Technik GmbH & Co. KG, Pforzheim, Germany) to produce either disc-shaped (10 mm in diameter and 3 mm in thickness, for microstructure analyses) or dumbbell-shaped specimens ($n = 6$, for tensile test according to ISO 22674 [17]). All the milled specimens were post-sintered in a furnace (SinTagon, Denstar, Daegu, Korea) under an argon gas purge at 1250, 1350 or 1450 °C for 1 h (a total of three groups) and finally subjected to natural cooling to room temperature.

### 2.2. X-ray Diffractometry (XRD) Analysis

The disc-shaped specimen surfaces were polished with silicon carbide papers and then finally with a 1-µm diamond suspension. Phase identification was carried out by X-ray diffractometry

(XRD, MAXima_X XRD-7000, Shimadzu, Kyoto, Japan), using Cu K$_\alpha$ radiation ($\lambda$ = 0.5418 nm) at an accelerating voltage of 30 kV, a beam current of 30 mA, a 2$\theta$-angle scan range of 20° to 100°, a scanning speed of 0.5°/min, a sampling pitch of 0.02°, and a preset time of 0.6 s. For comparison, a sample in the as-pre-sintered state was also subjected to XRD analysis. The phases were identified using spectra of known phases from the Joint Committee of Powder Diffraction Standard (JCPDS) database.

### 2.3. Microscopic Characterization

A specimen for each group was examined using field emission-scanning electron microscopy (FE-SEM, Merlin, Carl Zeiss AG, Oberkochen, Germany) under an accelerating voltage of 20 kV. The specimens were etched using the electrolytic etching method prior to SEM analysis. To determine the crystallographic orientation, electron backscattered diffraction (EBSD) scans were also performed on the FE-SEM equipped with a NordlysNano EBSD detector (Oxford Instruments, Abingdon, UK). A step size of 0.1 μm was used in a hexagonal scan grid.

### 2.4. Shrinkage and Density Analyses

To determine the shrinkage, the diameter and thickness of the disc-shaped specimens were measured before and after the sintering ($n$ = 3), and the percentage of the diameter and thickness reduction was then calculated as follows [14]:

$$\text{Percent of diameter shrinkage} = [(d_o - d_f)/d_o] \times 100\% \tag{1}$$

And

$$\text{Percent of thickness shrinkage} = [(h_o - h_f)/h_o] \times 100\% \tag{2}$$

where $d_o$ = diameter before sintering (mm), $d_f$ = diameter after sintering (mm), $h_o$ = height before sintering (mm) and $h_f$ = height after sintering (mm). The density of the post-sintered specimen was determined based on the Archimedes principle using a density determination kit (VPG214CN, Ohaus Corp., Parsippany, NJ, USA) at room temperature ($n$ = 3).

### 2.5. Tensile Test

The dumbbell-shaped specimens were loaded in tension (crosshead speed = 1.5 mm/min) using a universal testing machine (3366, Instron Inc., Canton, MA, USA), and four mechanical property values (ultimate tensile strength, 0.2% yield strength, percent elongation after fracture and Young's modulus) were calculated [17]. The fractured surfaces were examined with the FE-SEM.

### 2.6. Statistical Analysis

The mechanical properties were statistically analyzed with one-way analysis of variance (ANOVA) and Tukey's *post hoc* test ($\alpha$ = 0.05). The Young's modulus results were $\log_{10}$ transformed to satisfy homogeneity of variance before statistical analysis. The statistical analyses were performed with SPSS 17.0 for Windows (SPSS Inc., Chicago, IL, USA).

## 3. Results

### 3.1. XRD Analysis

Figure 1 shows the XRD patterns of the Co–Cr alloys in the as-pre-sintered state and in the three different post-sintered states. The as-pre-sintered sample showed a high peak intensity of $\gamma$ (face-centered cubic) phase and a low signal/noise ratio, indicating the initial densification during pre-sintering. All the three alloys basically had $\gamma$ and $\varepsilon$ (hexagonal close-packed) matrix phases. The 1250 °C group showed a higher peak intensity in the $\varepsilon$ phase than did the other two groups. In the 1350 °C and 1450 °C groups, the peak intensity of the $\gamma$ phase increased.

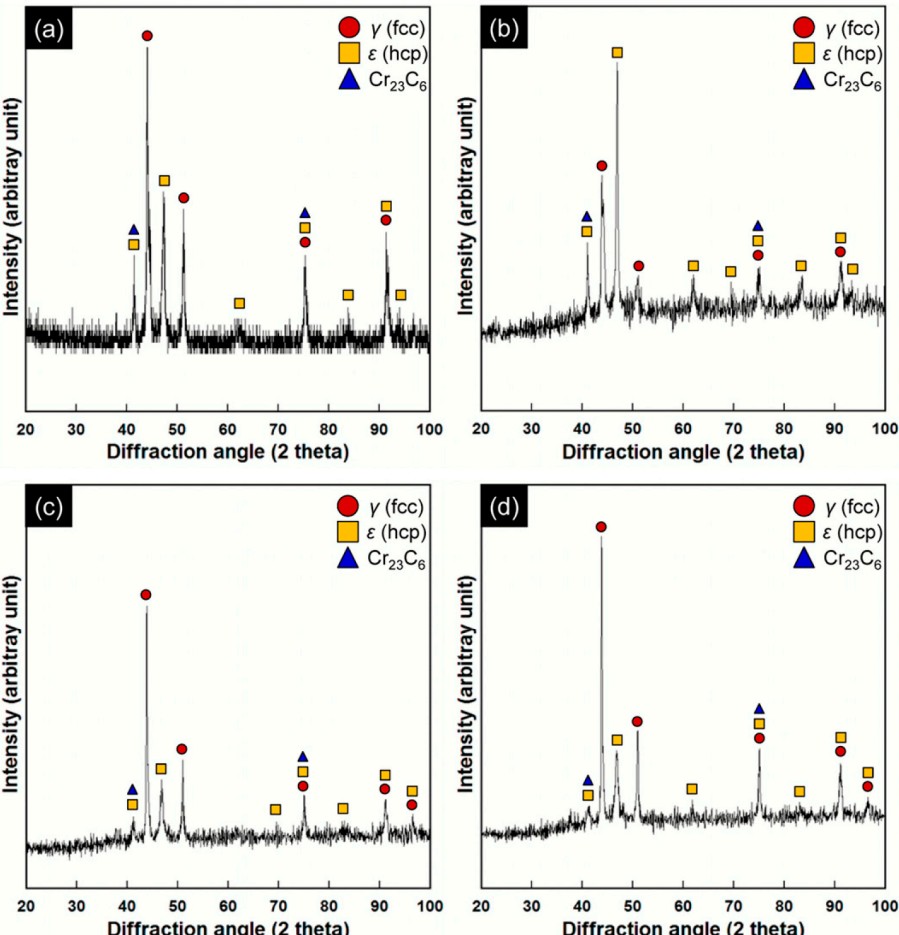

**Figure 1.** XRD patterns of the Co–Cr alloys in the as-pre-sintered state (**a**) and in the post-sintered states (prepared at three different temperatures): (**b**) 1250 °C; (**c**) 1350 °C; (**d**) 1450 °C. The Co-based $\gamma$ (face-centered cubic, fcc) and $\varepsilon$ (hexagonal close-packed, hcp) matrix phases were identified with JCPDS cards no. 15-806 and no. 05-727, respectively. The peaks indexed as $Cr_{23}C_6$ metal carbides were identified by JCPDS card no. 35-783.

*3.2. Microscopic Characterization*

Figure 2 presents the SEM images of the three different Co–Cr alloys. The corresponding phase and inverse pole figure (IPF) maps are also given in Figure 2. The formation of round-shaped pores was observed in all the SEM images. In particular, the 1250 °C group showed more pore formation than the other groups, with the number and size of the pores decreasing as the sintering temperature increased. The EBSD analysis results were consistent with those of the XRD analysis (Figure 1). All three groups had two matrix phases ($\gamma$ and $\varepsilon$ phases). In the 1250 °C group, however, the formation of the $\varepsilon$ phase was more predominant than that of the $\gamma$ phase, with little $Cr_{23}C_6$ carbide formation. Such carbide formation increased with a higher post-sintering temperature (within the grains in the 1350 °C group; along the grain boundaries in the 1450 °C group). The grain size of the 1350 °C group was slightly smaller than that of the 1250 °C group. However, the 1450 °C group showed the largest grain size among the three groups. In addition, the IPF maps revealed a significant number of annealing twins inside the matrix phases only in the 1450 °C group.

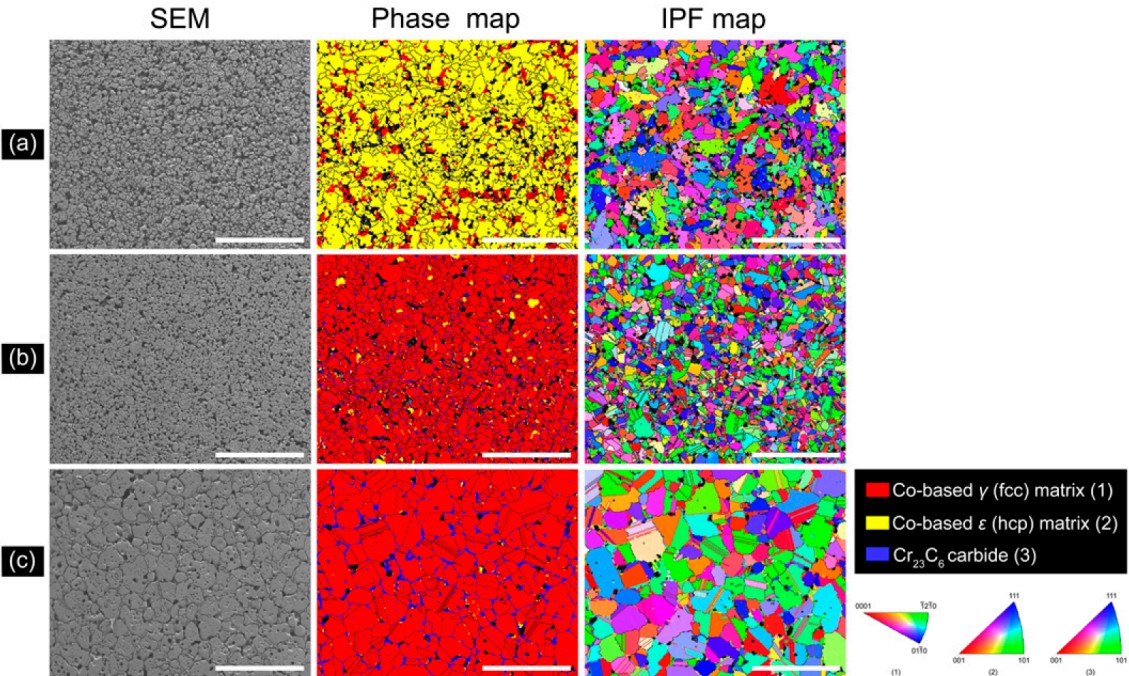

**Figure 2.** SEM/EBSD images of the Co–Cr alloys prepared at three different post-sintering temperatures: (**a**) 1250 °C; (**b**) 1350 °C; (**c**) 1450 °C (250×, scale bar = 100 μm).

### 3.3. Shrinkage and Density

Figure 3 exhibits the results of the shrinkage and density analyses for the Co–Cr alloys prepared at three different post-sintering temperatures. Both the thickness and the diameter of the alloys shrank significantly (approximately 7% to 13%) after post-sintering. Such post-sintering shrinkage increased with a higher post-sintering temperature, as did the density of the alloys.

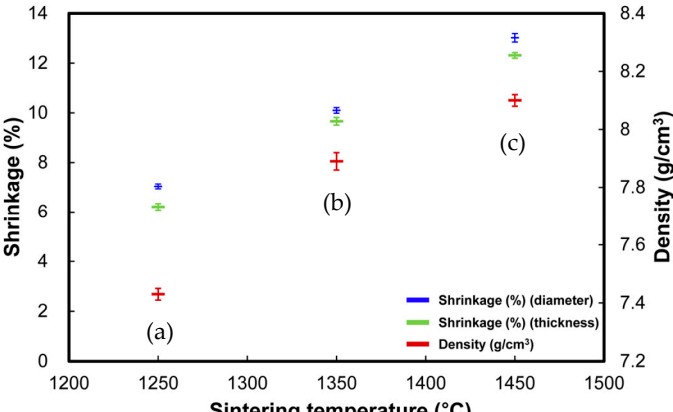

**Figure 3.** Percent of shrinkage and density of the Co–Cr alloys at three different post-sintering temperatures: (**a**) 1250 °C; (**b**) 1350 °C; (**c**) 1450 °C.

### 3.4. Mechanical Properties and Fractured Surfaces

Figure 4 presents the four mechanical properties of the three different Co–Cr alloys. In general, the 1350 °C group showed superior mechanical properties to the 1250 °C and 1450 °C groups. The 1350 °C group exhibited the highest mean ultimate tensile strength, followed by the 1450 °C group and then the 1250 °C group. The three groups are arranged in decreasing order of mean yield strengths (in MPa) as follows: 1350 °C (600 ± 23), 1450 °C (553 ± 31) and 1250 °C (492 ± 19). The mean percent elongation

values were substantially higher in the 1350 °C group (19 ± 2) than in the 1450 °C (10 ± 2) and 1250 °C (4 ± 1) groups. The mean Young's modulus values for all three groups were above 150 GPa.

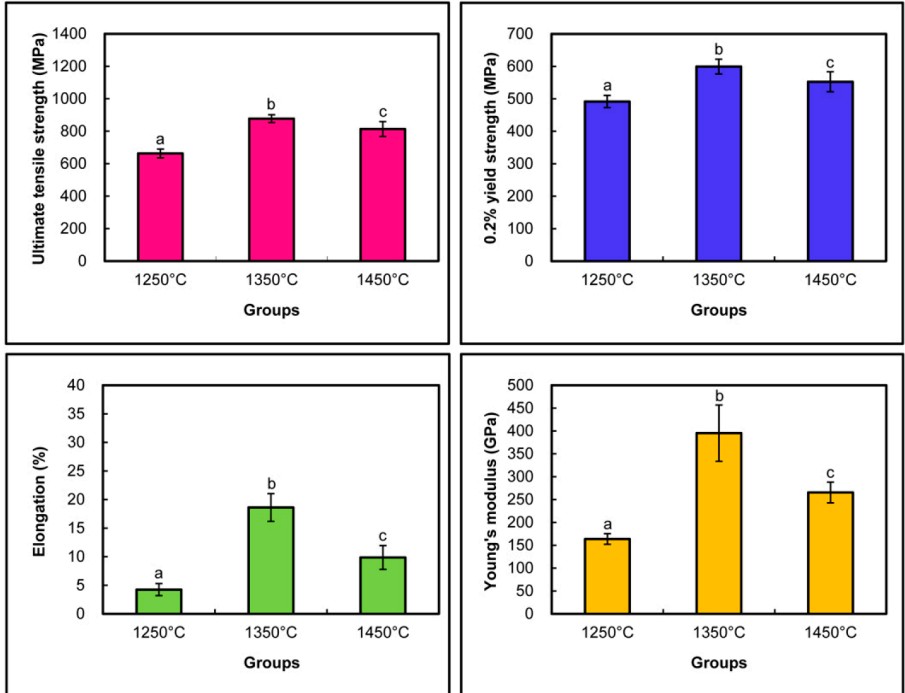

**Figure 4.** Comparison of mechanical properties of the Co–Cr alloys prepared at three different post-sintering temperatures (1250, 1350 and 1450 °C) (*n* = 6). For each figure, means with different lower-case letters indicate statistical differences among the three groups (*p* < 0.05).

The representative SEM images of the fractured surfaces are shown in Figure 5. All the groups clearly exhibited round-shaped pores on the fractured surfaces. The SEM images of the fractured surfaces also indicated a different microstructure for each Co–Cr alloy material depending on the post-sintering temperatures. The fractured surfaces of all the groups possessed lacerated ridges and dimples, indicating ductile tearing.

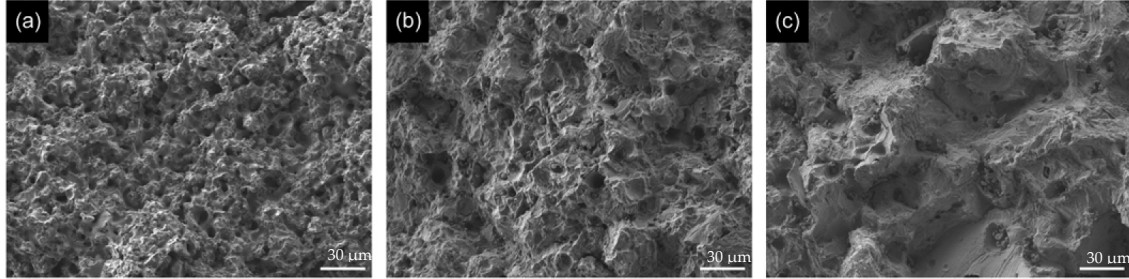

**Figure 5.** SEM image of the fractured surfaces (after tensile test) of the Co–Cr alloys prepared at three different post-sintering temperatures: (**a**) 1250 °C; (**b**) 1350 °C; (**c**) 1450 °C (500×, scale bar = 30 μm).

## 4. Discussion

The current study investigated the microstructures and mechanical properties of Co–Cr alloys post-sintered from a pre-sintered metal block at three different temperatures. The study results clearly showed that the microstructures of the post-sintered alloys were greatly dependent on the post-sintering temperature. In general, the 1350 °C group exhibited superior mechanical properties to the 1250 °C and 1450 °C groups. Therefore, the null hypothesis that there would be no significant

differences in the microstructures and mechanical properties among the alloys prepared with three different post-sintering temperatures was rejected.

The SEM images (Figure 2) showed that the low post-sintering temperature (1250 °C) resulted in low densification, with the original shape of the metallic particles still slightly remaining, as also seen in the low density value of the alloy (Figure 3). When the post-sintering temperature was increased to 1350 or 1450 °C, necking and bonding between the metallic particles increased, thereby producing more densified structures with fewer pores. The 1450 °C group showed grain growth (or coarsening) together with densification during post-sintering. Thus, increasing the post-sintering temperature resulted in greater shrinkage of the metallic bodies and, therefore, denser structures due to reduced porosity (Figures 1 and 3) [16]. These results are generally comparable to the previous findings of Kamardan et al. [14] and Wahi et al. [16].

Co is known to undergo an allotropic phase transformation from a high temperature fcc ($\gamma$) phase to a low temperature hcp ($\varepsilon$) phase [18]. The XRD and SEM/EBSD analysis results (Figures 1 and 2) indicated that all the three alloys consisted of $\gamma$ and $\varepsilon$ phases. In the 1350°C and 1450°C groups, the formation of the $\gamma$ phase was more predominant than that of the $\varepsilon$ phase, probably because a majority of the $\gamma$ phase was retained due to more rapid cooling [19–21]. In addition, the 1450°C group showed a higher peak intensity of the $\gamma$ phase than the 1350 °C group, indicating greater grain growth and enhanced crystallographic orientation [22]. The retained fcc structure, which was predominant in the 1350 °C and 1450 °C alloys, is believed to enhance the mechanical properties of the alloys [8,23]. In the 1250°C group, on the contrary, most matrix phases consisted of the $\varepsilon$ phase probably because less rapid cooling facilitated $\varepsilon$-martensite nucleation at a temperature range of 800–875 °C by isothermal aging [3,8,24–26]. In addition, it seems that the recrystallization of the pre-sintered Co–Cr alloy in which the $\gamma$ phase was predominant (Figure 2a) at the post-sintering temperature of 1250°C contributed to the formation of the $\varepsilon$ phase [27].

The mechanical properties of the Co–Cr alloys produced at three different post-sintering temperatures clearly showed that 1350 °C was the most optimal post-sintering temperature of the metallic blank (Figure 4). Unlike the 1250 °C group, the 1350°C group was mainly composed of the $\gamma$ phase rather than the $\varepsilon$ phase (Figures 1 and 2), yielding greater ductility of the alloy [28]. Nonetheless, the mechanical properties of the alloy were still superior to the 1250 °C group, probably due to the greater precipitation of $Cr_{23}C_6$ carbides within the grains [8,29]. Carbides can be precipitated in crystal defects such as dislocations and $\gamma/\varepsilon$ phase boundaries, such carbides in the grain interior are responsible for a dispersed strengthening effect of the alloy [29]. Grain boundaries also provide sites for the preferential precipitation of carbides. However, grain boundary carbides can cause low ductility and are detrimental to the fatigue strength of the alloys [29]. The 1450 °C group showed a more brittle character than the 1350 °C group, probably due to carbide formation along the grain boundaries and the twin boundary formation (Figure 2c) [8,29]. In addition, the 1450 °C group exhibited a larger grain size due to prolonged crystal growth at a higher post-sintering temperature. Such a larger grain size may be detrimental to the desired mechanical strength (in particular, yield strength and ductility) according to the Hall–Petch relationship [30,31]. According to ISO 22674 [17], the mechanical properties of the 1350 °C and 1450 °C alloys satisfied the Type 5 criteria and, therefore, can be widely used for the fabrication of any type of dental metallic prostheses (from single-tooth fixed restorations to thin removable partial dentures, parts with thin cross-sections and clasps). The 1250 °C alloy showed slightly inferior mechanical properties and did not satisfy the Type 5 criteria, indicating that its use on thin dental structures should be avoided.

In this study, un-sintered and newly-developed, pre-sintered Co–Cr metallic blocks were not evaluated together. Therefore, the potential laboratory and clinical merits of a pre-sintered block over an un-sintered block should be further investigated. The post-sintering temperature can also affect the fitting accuracy of the metallic restoration by altering post-sintering shrinkage (Figure 3) [13]. When using soft Co–Cr metallic blocks, therefore, one must consider the amount of shrinkage after sintering during the CAM step [13]. The findings of this in vitro study suggest that the mechanical properties

of pre-sintered Co–Cr alloys are significantly altered by the post-sintering temperature. However, a variety of factors should be further optimized to increase the efficacy in the production of metallic dental restorations using the alloy system.

## 5. Conclusions

In this study, Co–Cr alloys were post-sintered from a pre-sintered metal block (soft metal) at three different temperatures (1250, 1350 and 1450 °C) and their microstructures and mechanical properties were compared. The microstructures of the post-sintered alloys were greatly dependent on the post-sintering temperature. In terms of mechanical properties, 1350 °C was found to be the most optimal among the three post-sintering temperatures. It seems that over-sintering showed better mechanical properties than under-sintering during the post-sintering stage.

**Author Contributions:** Conceptualization, S.-H.J., M.-H.H. and T.-Y.K.; methodology, B.K.M. and M.-H.H.; validation, B.K.M. and T.-Y.K.; formal analysis, T.-Y.K.; investigation, S.-H.J., M.-H.H. and T.-Y.K.; resources, T.-Y.K.; writing—original draft preparation, S.-H.J., M.-H.H. and T.-Y.K.; writing—review and editing, T.-Y.K.; visualization, S.-H.J. and M.-H.H.; supervision, B.K.M. and T.-Y.K.; project administration, T.-Y.K.; funding acquisition, M.-H.H.

**Funding:** This research was supported by the Basic Science Research Program through the National Research Foundation of Korea (NRF) funded by the Ministry of Education (NRF-2017R1A6A3A11036498).

**Conflicts of Interest:** The authors declare no conflict of interest. The funders had no role in the design of the study; in the collection, analyses, or interpretation of data; in the writing of the manuscript, or in the decision to publish the results.

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
