# Peer review of "Effect of Different Post-Sintering Temperatures on the Microstructures and Mechanical Properties of a Pre-Sintered Co–Cr Alloy"

_metals, doi:10.3390/met8121036_

Reviewer 1 Report

Dear Authors,

.        The topic of this paper – Effect of Different Post-Sintering Temperatures on the Microstructures and Mechanical Properties of a 
 Pre-Sintered Co–Cr Alloy 
- is of high current relevance and fits the aim of the journal. I have read your manuscript with great interests and I have some questions and suggestions that I would like for you to address:

1.    Pasge 3; 2.1. Specimen Preparation, The cooling condition should be mentioned because phase change is strongly affected by cooling rate. 

2.    Page4; 2.4. Shrinkage and Density Analyses, The unit for the length might be better in “mm” rather than “cm” if there is no particular reason because the specimen size is written in “mm”. 

3.    Page4, The XRD Patterns of as pre-sintered condition should be added for well understanding of phase change during post-sintering procedure.

4.    Page 5; Figure 2, The inverse pole figures of fcc and hcp are wrong. Those must be corrected.

5.    Page 5; Figure 3, The independent variable should be described in horizontal axis while dependent variable should be in vertical axis.  

6.    Page6; Figure 4., The group name in the graph is wrong. Those must be corrected.

7.   Page7, In discussion, that the epsilon phase observed more in 1250°C 
is due to less rapid cooling. Why cooling rate in 1250°C is less than other conditions?

 Considering the chemical composition of Co-Cr alloy used in this study, the γ phase should be dominant in 1250 °C 
based on the phase diagram. In addition, usually more pore increase the cooling rate due to its more surface area. Thus the γ phase could be the dominant phase in 1250°C. However the results are not like so. The rational explanation and deep discussion are needed why the epsilon phase are dominant phase in 1250°C samples. In addition Reference 13 does not mention about the phase at all. Is this reference suitable for this part?

Author Response

The topic of this paper – Effect of Different Post-Sintering Temperatures on the Microstructures and Mechanical Properties of a Pre-Sintered Co–Cr Alloy - is of high current relevance and fits the aim of the journal. I have read your manuscript with great interests and I have some questions and suggestions that I would like for you to address:

1.    Pasge 3; 2.1. Specimen Preparation, The cooling condition should be mentioned because phase change is strongly affected by cooling rate. 

Answer: Thank you for your suggestion. We have added the cooling rate information in the subsection.

2.    Page4; 2.4. Shrinkage and Density Analyses, The unit for the length might be better in “mm” rather than “cm” if there is no particular reason because the specimen size is written in “mm”. 

Answer: We agree with your suggestion. We have changed the unit to mm in the subsection.

3.    Page4, The XRD Patterns of as pre-sintered condition should be added for well understanding of phase change during post-sintering procedure.

Answer: We have newly added the XRD spectrum of a sample in pre-sintered condition (Figure 1), as you suggested.

4.    Page 5; Figure 2, The inverse pole figures of fcc and hcp are wrong. Those must be corrected.

Answer: Thank you for pointing it out. We have carefully modified Figure 2, as you commented.

5.    Page 5; Figure 3, The independent variable should be described in horizontal axis while dependent variable should be in vertical axis.  

Answer: We agree with your comment. We have modified Figure 3, as you suggested.

6.    Page6; Figure 4., The group name in the graph is wrong. Those must be corrected.

Answer: Thank you very much for pointing it out. This careless mistake has been corrected.

7.   Page7, In discussion, that the epsilon phase observed more in 1250°C is due to less rapid cooling. Why cooling rate in 1250°C is less than other conditions? Considering the chemical composition of Co-Cr alloy used in this study, the γ phase should be dominant in 1250 °C based on the phase diagram. In addition, usually more pore increase the cooling rate due to its more surface area. Thus the γ phase could be the dominant phase in 1250°C. However the results are not like so. The rational explanation and deep discussion are needed why the epsilon phase are dominant phase in 1250°C samples. In addition Reference 13 does not mention about the phase at all. Is this reference suitable for this part?

Answer: We have revised the paragraph to answer your valuable questions. It does not seem that the performance of the Co-Cr powder metallurgy alloy used in our study is necessarily consistent with the phase diagram. Once again, we truly appreciate all your careful reading and comments.

Reviewer 2 Report

Very interesting work on Co-Cr alloys for prosthetic dentistry. The following points need to be clarified:

1- The loss of yield point produced by oversintering is associated to Cr23C6 precipitation at grain boundaries (comparing the results obtained at 1300ºC and 1400ºC). Is there any effect of grain growth? Hall Petch effect should be taken into account

2- Identification of Cr23C6 carbide by XRD (Fig. 1) is not conclusive. It is better identified in EBSD images.

3- Fractographic analysis is needed in order to identify crack initiation flaws and confirm that these are carbides.

4- Is transformation induced plasticity possible in these alloys? Is FCC stable enough or does it transform to HCP during fracture?

Author Response

Very interesting work on Co-Cr alloys for prosthetic dentistry. The following points need to be clarified:

1- The loss of yield point produced by oversintering is associated to Cr23C6 precipitation at grain boundaries (comparing the results obtained at 1300ºC and 1400ºC). Is there any effect of grain growth? Hall Petch effect should be taken into account

Answer: Thank you for your valuable comment. We have revised the related sentences in the Discussion section.

2- Identification of Cr23C6 carbide by XRD (Fig. 1) is not conclusive. It is better identified in EBSD images.

Answer: We agree with your suggestion. We have modified the description of the XRD analysis in the Results section.

3- Fractographic analysis is needed in order to identify crack initiation flaws and confirm that these are carbides.

Answer: Thank you for your suggestion. Unfortunately, however, samples for further fractographic analysis are not currently available. The formation of carbides within the alloys was confirmed by the EBSD analysis (Figure 2).

4- Is transformation induced plasticity possible in these alloys? Is FCC stable enough or does it transform to HCP during fracture?

Answer: Thank you again for all your valuable comments and suggestions. The transformation could occur during fracture (even during polishing). In our study, however, such potential transformation during plastic deformation was not studied directly.